# Anti-Pectin Fouling Performance of Dopamine and (3-Aminopropy) Triethoxysilane-Coated PVDF Ultrafiltration Membrane

**DOI:** 10.3390/membranes12080740

**Published:** 2022-07-28

**Authors:** Dengrong Lu, Hongbo Liu, Zhishu Tang, Mei Wang, Zhongxing Song, Huaxu Zhu, Dawei Qian, Xinbo Shi, Guolong Li, Bo Li

**Affiliations:** 1Co-Construction Collaborative Innovation Center for Chinese Medicine Resources Industrialization by Shaanxi & Education Ministry, Shaanxi University of Chinese Medicine, Xianyang 712038, China; ludengrongg@163.com (D.L.); wangm1379@163.com (M.W.); 13992092403@163.com (Z.S.); 1501026@sntcm.edu.cn (X.S.); weercog@126.com (G.L.); 2State Key Laboratory of Research & Development of Characteristic Qin Medicine Resources (Cultivation), Shaanxi University of Chinese Medicine, Xianyang 712038, China; 3China Academy of Chinese Medical Sciences, Beijing 100700, China; 4Wang Jing Hospital of China Academy of Chinese Medical Sciences, Beijing 100102, China; 5Jiangsu Collaborative Innovation Center of Chinese Medicinal Resources Industrialization, Nanjing University of Chinese Medicine, Nanjing 210023, China; zhuhx@njucm.edu.cn (H.Z.); qiandwnj@126.com (D.Q.); boli@njucm.edu.cn (B.L.); 6Jiangsu Botanical Medicine Refinement Engineering Research Center, Nanjing University of Chinese Medicine, Nanjing 210023, China

**Keywords:** ultrafiltration membrane, polydopamine, (3-Aminopropy) triethoxysilane, antifouling, pectin, XDLVO theory

## Abstract

Due to the diversity and complexity of the components in traditional Chinese medicine (TCM) extracts, serious membrane fouling has become an obstacle that limits the application of membrane technology in TCM. Pectin, a heteropolysaccharide widely existing in plant cells, is the main membrane-fouling substance in TCM extracts. In this study, a hydrophilic hybrid coating was constructed on the surface of a polyvinylidene fluoride (PVDF) ultrafiltration (UF) membrane co-deposited with polydopamine (pDA) and (3-Aminopropy) triethoxysilane (KH550) for pectin antifouling. Characterization analysis showed that hydrophilic coating containing hydrophilic groups (–NH_3_, Si-OH, Si-O-Si) formed on the surface of the modified membrane. Membrane filtration experiments showed that, compared with a matched group (FRR: 28.66%, Rr: 26.87%), both the flux recovery rate (FRR) and reversible pollution rate (Rr) of the pDA and KH550 coated membrane (FRR: 48.07%, Rr: 44.46%) increased, indicating that pectin absorbed on the surface of membranes was more easily removed. Based on the extended Derjaguin–Laudau–Verwey–Overbeek (XDLVO) theory, the fouling mechanism of a PVDF UF membrane caused by pectin was analyzed. It was found that, compared with the pristine membrane (144.21 kT), there was a stronger repulsive energy barrier (3572.58 kT) to confront the mutual adsorption between the coated membrane and pectin molecule. The total interface between the modified membrane and the pectin molecule was significantly greater than the pristine membrane. Therefore, as the repulsion between them was enhanced, pectin molecules were not easily adsorbed on the surface of the coated membrane.

## 1. Introduction

Membrane technology is an efficient separation technique. Many researchers are attracted to the technology due to its unique separation principle, i.e., selective transport and efficient separation [1]. In recent years, membrane technology has been widely used for the separation, purification, and concentration of traditional Chinese medicine (TCM). However, the TCM system is extremely complex. In addition to the active ingredients, some macromolecular substances, such as pectin, starch, and tannin, are commonly present in TCM extracts. These substances have little or no pharmacological activity, resulting in large doses, and making it difficult to prepare TCMs. These substances usually need to be removed by refining. However, when separating them using membrane technology, these substances are easily adsorbed on the surface of the membrane, around the membrane pores, and on the inner wall, causing serious membrane fouling.

Structurally and functionally, pectin is the most complex polysaccharide in plant cell walls [2]. It is composed of 500 to 1000 galacturonic acid residues. Pectin easily leads to a negatively charged colloidal solution dissolved in water [3]. This will agglomerate to varying degrees according to the chemical structure, solution environment, and interaction within molecules. In membrane experiments with TCM, substances were absorbed on the surface of the membrane or inside the pores, causing reversible or irreversible fouling. Due to its structural complexity and gel property, pectin is one of the primary foulants that causes membrane fouling in TCM membrane experiments. In addition, previous studies on the membrane fouling of four macromolecules—pectin, starch, tannin, and protein—illustrate that pectin easily causes more serious fouling of UF membranes [4].

Membrane fouling results from the complicated interaction between the membrane and foulant during the filtration operation. This is related to the properties of the membrane, and to the feed system and operational conditions. Foulants are deposited on the membrane surface and around the holes, and adhere to the inner wall of membrane holes. The polyvinylidene fluoride (PVDF) membrane [5,6,7] is a synthetic polymer membrane that more easily absorbs organic molecules on the surface during the TCM separation process due to its inherent hydrophobicity.

With the application and popularization of membrane technology, membrane fouling is the primary problem that needs to be solved. Three major factors affect membrane fouling: the solution environment [8,9], operating conditions, and membrane properties. As TCM water extract is a complex system, changing the solution environment involves adding some acid or base to adjust the pH, or adding Na^+^, Ca^2+^, and Mg^2+^ to adjust the ionic strength and influence the active ingredients. This involves modifying the membrane to change the physical properties of the surface, such as hydrophilia, electrical and adsorption activity, to adjust the interaction between the membrane and foulants, reducing membrane fouling [10,11]. Recently, dopamine has been viewed as a promising bio-inspired material for membrane modifications. Due to the abundant catechol groups and ethylamine groups in dopamine [12,13,14], a hydrophilic polymer layer can be coated on the surface of many membrane materials, such as PVDF, polyamide (PA), and polyacrylonitrile (PAN), through the oxidation–polymerization of dopamine [15,16]. Due to the limited hydrophilicity and instability of acidic, basic, and organic solvents, precursors of hydrotropic substances such as tetraethyl orthosilicate (TEOS) [17], silane coupling agent (KH550, KH560, KH570, et al.) [18,19], poly (ethylene imine) [20], and ammonium fluorotitanate [15] were added to dopamine solutions, as shown in Table 1. The hydrolysis of the substance was synchronized with the polymerization of dopamine and absorbed the hydrophilic functional group on the surface of membrane. This not only greatly enhanced the hydrophilic properties of the membrane and optimized the membrane performance, but also improved the stability of the modified coating through the interaction between substances.

This paper combined the oxidative polymerization of dopamine and the hydrolysis of KH550. The polydopamine coating on the membrane surface fixed the KH550 hydrolysis product on the surface by means of hydrogen bonding and physical cross-linking winding, forming a hybrid coating, as shown in Figure 1. Pectin, as a model foulant, was used to investigate the performance of modified membranes. The surface of the coated membrane was characterized by scanning electron microscopy (SEM), energy dispersive X-ray spectroscopy (EDS), attenuated total internal reflectance fourier transform infrared spectroscopy (ATR-FTIR) and contact angle analysis. The anti-fouling performance of the modified membrane was evaluated by using a pectin solution as a model foulant, and the mechanism of pollution was analyzed from a microscopic point of view by the extended Derjaguin–Laudau–Verwey–Overbeek (XDLVO) theory. To provide ideas for the prevention and control of membrane pollution, a relatively complete and feasible evaluation system for membrane modification was established.

## 2. Materials and Methods

### 2.1. Materials

A flat PVDF UF membrane with a molecular weight cut-off of 50 kDa was provided by Rising Sun Membrane Technology Co. Ltd. (Beijing, China). Pectin, dopamine hydrochloride (pDA), and 3-Aminopropy triethoxysilane (KH550) were all purchased from Macklin Biochemical Co. Ltd. (Shanghai, China).

### 2.2. Membrane Preparation

Before modifying the membrane, the PVDF UF membrane is pretreated [21,23] as follows: the membrane was soaked in purified water for 12 h to remove glycerol on the surface of the membrane, soaked in 95% ethanol for 5 min to ensure the membrane was completely moistened, soaked in purified water for 12 h to remove all ethanol, then was stored in pure water. A certain amount of dopamine hydrochloride was dissolved in Tris-HCl (pH 8.5) solution, and the silane coupling agent KH550 was added to part of it, then placed on a magnetic stirrer (25 °C, 500 r/min) and stirred for 0.5 h with a magnetic stirrer (25 °C, 500 r/min) to form the modified solution. The pretreated PVDF ultrafiltration membrane was placed in a glass Petri dish with an appropriate amount of modified liquid. This was oscillated at a constant temperature (25 °C, 40 r/min) under aerobic conditions for 8 h, and at the end of the reaction, the excess particles and unreacted dopamine were washed off with deionized water. Finally, the modified membrane was obtained and placed in deionized water for storage.

### 2.3. Membrane Characterization

A dynamic contact angle meter (DSA100S, Hamburg, Germany) was used to measure the static liquid contact angle of the membrane, the cake layer of pectin and the corresponding surface energy, to examine the membrane surface wettability. In order to more intuitively reflect the changes in the hydrophilicity of the membrane surface, the dynamic contact angle of deionized water on the surface of each sample membrane was measured every ten seconds. A zeta-potential analyzer (Surpass, Graz, Austria) was used to determine the zeta potential of the PVDF ultrafiltration membrane and various polymer contaminants. The test solution was KCl solution (1 mol/L), and the elements and morphologies of the surface of the located and control membranes were tested using energy dispersive X-ray spectroscopy (EDS, Xplore, Oxford, UK), scanning electron microscopy (SEM, TESCAN MIRA4, CZE), and Fourier transform infrared spectroscopy (FTIR) with attenuated total reflection (ATR).

The overall porosity (ω) and the mean pore radius ($r_{m}$) were, respectively, calculated by Formulas (1) and (2). The overall porosity was determined by the gravimetric method. The membrane mean pore radius was determined on the basis of the pure water and porosity data [24,25].
(1)$$\omega \;\left(\%\right)=\frac{w_{1}-w_{2}}{A\times l\times ∆P}$$
(2)$$r_{m}=\sqrt{\frac{\left(2.9-1.75\times \omega \right)\times 8\times \eta \times l\times J_{w}}{\omega \times A\times ∆P}}$$
where $w_{1,\;}w_{2}$ are wet and dry weights of the membrane (g), respectively; A is the membrane effective area (0.004 m^−2^); l is the membrane thinkness (m); $d_{w}$ is the water density ($9.98\times 10^{5}$ g·m^−3^); η is the water viscosity ($8.9\times 10^{-4}$ pa·s); $J_{w}$ is the permeated pure water amount per unit time (m^3^·s^−1^), and ∆P is the operation pressure (200 kPa).

### 2.4. Filtration Experiments

The pretreated PVDF membrane and the modified membrane were compacted at 300 kPa for 10 min to obtain a steady water flux. Dead-end equipment (Millipore, XFUF07601) was used for filtration. The next operation steps were as follows: (1) pure water was used to filter at 200 kPa pressure for 1 h, and the flux was recorded at 200 kPa every 1 min to collect five steady readings to obtain an average value (*J_W0_*); (2) the flux of pectin solution (*J_f_*) was recorded for 3 h under the same conditions and by adjusting the speed of the magnetic stirrer to 180 r/min; (3) the water flux (*J_w1_*) of the fouled membrane was measured for 0.5 h at the same pressure as in step (1); (4) the pectin-fouled membranes were washed with pure water for about 0.5 h and then inverted in the ultrafiltration device and cleaned for 0.5 h at 200 kPa, after which the water flux of the cleaned membranes (*J_W2_*) was measured again.

All flux and pectin rejection values were calculated by the following formulas [19,26]:(3)$$J=\frac{V}{A\cdot t}\;\;$$
(4)$$R\left(\%\right)=\left(1-\frac{C_{P}}{C_{F}}\right)\times 100\%\;$$
where J is the permeate flux (L·m^−2^·h^−1^); V is the permeate volume measured (L); A is the membrane active area (m^2^); t is the permeation time (h); R (%) is the pectin rejection; and $C_{P}$ and $C_{F}$ are the concentrations of pectin in the permeation and feed solution, respectively.

The anti-fouling performance of the membrane was evaluated by the recovery rate of the flux and the fouling rate. The recovery rate of the flux and the proportion of reversible fouling to total fouling were positively correlated with the anti-fouling performance of the membrane [27,28]:(5)$$FRR=\frac{J_{W2}}{J_{W0}}\times 100\%\;$$
(6)$$Rt=\left(1-\frac{J_{f}}{J_{W0}}\right)\times 100\%\;$$
(7)$$Rr=\left(\frac{J_{W2}-J_{f}}{J_{W0}}\right)\times 100\%\;$$
(8)$$Rir=\left(\frac{J_{W0}-J_{W2}}{J_{W0}}\right)\times 100\%\;$$
where FRR is the recovery rate of flux; Rt is the total fouling rate; and Rr and  Rir  are the reversible fouling rate and irreversible fouling rate, respectively.

### 2.5. XDLVO

According to XDLVO theory, the total interface energy between foulant and membrane, including polar force (AB), van der Waals force (LW), and electric double layer force (EL) [29], was calculated as follows:(9)$$G_{mlp}^{TOT}=G_{mlp}^{AB}+G_{mlp}^{LW}+G_{mlp}^{EL}$$
where m, l and p are the membrane, liquid, and pectin, respectively. $G_{mlp}^{TOT}$ > 0 indicates that there is mutual attraction between membrane and foulant, and, in reverse, is mutually exclusive. In Equation (9), $G_{mlp}^{AB}$, $G_{mlp}^{EL}$ and $G_{mlp}^{LW}$ are calculated by the following formula [30,31,32]:(10)$$∆G_{mlp}^{AB}=2\sqrt{\gamma_{l}^{+}}\left(\sqrt{\gamma_{m}^{-}}+\sqrt{\gamma_{p}^{-}}+\sqrt{\gamma_{l}^{-}}\right)+2\sqrt{\gamma_{l}^{-}}\left(\sqrt{\gamma_{m}^{+}}+\sqrt{\gamma_{p}^{+}}+\sqrt{\gamma_{l}^{+}}\right)+2\left(\sqrt{\gamma_{m}^{+}\gamma_{p}^{-}}+\sqrt{\gamma_{m}^{-}\gamma_{p}^{+}}\right)$$
(11)$$∆G_{mlp}^{LW}=2\left(\sqrt{\gamma_{l}^{LW}}-\sqrt{\gamma_{m}^{LW}}\right)\left(\sqrt{\gamma_{p}^{LW}}-\sqrt{\gamma_{l}^{LW}}\right)$$
(12)$$∆G_{mlp}^{EL}=\frac{\varepsilon_{0}\varepsilon_{r}k}{2}\cdot \left(\zeta_{m}^{2}+\zeta_{p}^{2}\right)\left[1-\coth\left(kd_{0}\right)+\frac{2\zeta_{m}\zeta_{p}}{\zeta_{m}^{2}\zeta_{p}^{2}}\mathrm{csch}\left(kd_{0}\right)\right]$$
where $\gamma^{LW}$, $\gamma^{+}$, $\gamma^{-}$ are the surface tension parameters of LW component, electron acceptor, and electron donor, respectively; $\varepsilon_{0}$ ($8.85\times 10^{-12}F/m$) is the permittivity of a vacuum; $\varepsilon_{r}$ (80) is the relative permittivity of the solution; k (0.104 nm^−1^) is the derivative of the Debye constant; $d_{0}$ (0.158 nm) is the minimum distance between the two planes; and $\;\zeta_{m}$, $\zeta_{p\;}$ is the zeta potential of the membrane surface and pectin filter cake layer, respectively.$\;\gamma_{l}^{+}$, $\gamma_{l}^{-}$, $\gamma_{l}^{LW}$ are known, and $\gamma_{m}^{+}$, $\gamma_{m}^{-}$, $\gamma_{m}^{LW}$, $\gamma_{p}^{+}$, $\gamma_{p}^{-}$, $\gamma_{p}^{LW}$ are, respectively, computed by combining known parameters and the Young equation [33,34,35].
(13)$$\gamma^{TOT}=\gamma^{LW}+\gamma^{AB}$$
(14)$$\gamma^{AB}=2{\left(\gamma^{-}\gamma^{+}\right)}^{1/2}\;$$
(15)$$\gamma_{l}\left(1+\cos\theta \right)=2\sqrt{\gamma_{s}^{LW}\gamma_{s}^{LW}}+2\sqrt{\gamma_{l}^{-}\gamma_{s}^{+}}+2\sqrt{\gamma_{s}^{-}\gamma_{l}^{+}}\;$$
where θ is the contact angle of the solution on the surface to be measured; lower corner l is liquid to be measured; and *s* is surface to be measured (membrane or cake layer of fouling).

In order to more accurately describe the interface energy between the membrane and foulants, the fouling was regarded as a spherical molecule and the membrane was regarded as an infinite plane. A certain functional relationship exists between the interface function and the interface distance. The final adhesion of a foulant on the membrane surface is determined by the short-ranged interaction energy between them, which can be quantified by methods provided in XDLVO theory. The strength of the energy between the membrane and the foulant at a certain separation distance was quantified by the following equations [29,30,36]:(16)$$U^{TOT}=U^{AB}+U^{LW}+U^{EL}$$
(17)$$U_{mlf}^{LW}\left(h\right)=2\pi d_{0}^{2}∆G_{mlf}^{LW}\left(\frac{a}{h}\right)\;$$
(18)$$U_{mlf}^{AB}\left(h\right)=2\pi a\lambda \Delta G_{mlf}^{AB}exp\left(\frac{d_{0}-h}{\lambda }\right)\;$$
(19)$$U_{mlf}^{EL}\left(h\right)=\pi a\varepsilon_{r}\varepsilon_{0}\left[2\zeta_{m}\zeta_{f}\ln\left(\frac{1+exp\left(-kh\right)}{1-exp\left(-kh\right)}\right)+\left(\zeta_{m}^{2}+\zeta_{f}^{2}\right)\ln\left(1-exp\left(-2kh\right)\right)\right]$$
where a is the hydraulic radius of the foulants and  λ (0.6 nm) is the characteristic length of the polar interaction in water.

## 3. Results and Discussion

### 3.1. Contact Angle, SEM, and ATR-FTIR Analyses

The color change and morphological variation of the membranes were tracked by SEM and photographs, as shown in Figure 2, respectively. In the pristine PVDF membrane, the whole membrane was white. From its corresponding SEM images, we observed that the surface was very flat and smooth, and there were many evenly distributed and irregularly shaped nanochannels on the surface of the pristine membrane. When coating DA on the surface of the PVDF membrane, the color became a little brown, part of the hole was covered, and some bulk polymers could be detected on the membrane surface. This was due to the dopamine forming polydopamine particles by oxidative self-polymerization on the membrane surface. On introducing KH550 into the system, the brown became darker and the surface was relatively flat and smooth compared with the membrane that was only coated with dopamine, without raised lumps on the surface of the membrane and only some heterozygous particles with a relatively uniform distribution. The alkoxy group of the silane coupling agent had a hydrogen bond with the phenolic hydroxyl group of the pDA after hydrolysis, and condensed with itself to form a silicon-containing oligomer [36]. This result was verified by the results of the EDS. As shown in Figure 3 and Table 2, elements such as carbon, nitrogen, oxide and fluoride were mainly detected on the surface of the pristine PVDF membrane, and the weight of carbon (60.59%) was the largest, followed by fluoride (38.38%). After the modification of pDA, the weight of carbon increased slightly and the fluorine content decreased significantly. In addition, nitrogen (3.05%) was also detected on the membrane surface. It indicated that the membrane surface was covered with a polydopamine coating. The introduction of KH550 not only increased the relative weight of nitrogen (3.30%), but also detected silicon with a weight of 0.51%. Combining the result of SEM and EDS, it was inferred that a coating containing silicon and nitrogen was formed on the membrane surface through complex cross-linking between pDA and KH550.

The surface contact angles of the pristine PVDF and pDA-coated PVDF membranes are shown in Figure 4. The water contact angle of the pristine PVDF membrane decreased from 82° to 68° after being modified by pDA for 8 h. It reduced further to 40° after the addition of KH550. The contact angle continuously decreased over time, dropping to 20° at 60 s. However, the contact angle of the pristine and only-dopamine-coated membrane did not change obviously over time. This shows that the hydrophilic coating successfully formed on the surface of the PVDF membrane after the modification pDA + KH550, which was further confirmed by ATR-FTIR. The ATR-FTIR spectra of the pristine and modified PVDFs are shown in Figure 5. After the pDA modification, several new absorption signals appeared. A broad absorbance between 3600 and 3100 cm^−1^ was ascribed to N–H/O–H stretching vibrations. A peak of 1605 cm^−1^ was attributed to the aromatic ring vibrations. In addition to the constant absorption peak, Si-OH (952 cm^−1^) and Si-O-Si (1075 cm^−1^) were detected on the coated membrane surface of pDA + KH550. The result of the analysis combining the contact angle and ATR-FTIR preliminarily indicate that the introduction of the hydrophilic functional group enhanced the hydrophilicity and permeability of the membrane surface [19].

### 3.2. Analysis of Anti-Fouling and Separation Performance

In order to investigate the performance of the coated membrane, a set of controlled trials were designed using pectin as a model foulant to detect the separation and anti-fouling performance of the membranes. The results are displayed in Figure 6. As shown in Figure 6a, the relative flux of the coated membranes using the pectin solutions was greater than that of the pristine membrane, indicating an improvement in anti-fouling after modification. Figure 6b is the pure water permeate flux of different membranes in various filtration stages as described in “2.4. Filtration Experiments”, including the water flux of initial membranes (J_W0_), fouled membranes (J_W1_), and cleaned membranes (J_W2_). The modified membrane developed a polymer coating on its surface, as seen by the SEM (Figure 2). Due to the membrane pores being blocked by the polymer covering, the initial water flux of the modified membrane was smaller than that of the pristine membrane. After filtration of the pectin solution, the water flux of all membranes was significantly reduced, mainly due to the adsorption of pectin molecules and the cake layer on the membrane surface. There was no significant difference in the water flux of all fouled membranes. After being washed with pure water, the cleaned membranes’ water flux of pDA + KH550-coated membranes is significantly higher than the pDA-coated membrane and pristine membrane. Combining the flux of water and pectin solution, Equations (5) – (8) were applied to calculate the fouling indicators, as shown in Figure 7. The reversible fouling rate and the recovery rate of the flux increased after the modification, and the pDA + KH550-coated membrane was superior to the pDA-coated membrane. It indicated that foulants adsorbed on the pDA + KH550-coated membranes were more easily removed, compared with the pristine membrane and pDA-coated membrane.

Figure 8 and Figure 9, respectively, illustrate pectin rejection by the pristine and coated membranes, and the particle size distribution of the pectin solution. Compared with the pristine film, the rejection of the coated membrane by dopamine increased slightly. After introducing KH550, the value increased significantly to 96.22%. As shown in Figure 9, the particle size distribution range of the pectin solution was wide. The reason for this was that pectin molecules in the colloidal solution easily condensed with each other and formed different sizes according to the degree of aggregation, resulting in an uneven molecular weight. The particle size distribution of the pectin molecules in the pectin permeating through the coating membrane were more concentrated, and the curve obviously shifts to the left, indicating that after passing the membrane, the pectin molecules in the solution that were greater than the membrane pore size were intercepted. Furthermore, the molecular size in the translucent solution was more uniform. As shown in Table 3, the mean pore size of the pristine PVDF membrane was 40.18 nm and the overall porosity was 53.11%. After modification, both the mean pore size and overall porosity were reduced due to the membrane surface being covered with a hydrophilic coating. This reduction was even more pronounced for the pDA + KH550 modified membrane, where the mean pore size decreased to 27.25 nm and the overall porosity fell to 43.69%. Therefore, for the pDA + KH550 modified membrane, more macromolecules were intercepted and the molecular size in the permeate was more uniform, as observed in Figure 9.

### 3.3. XDLVO Theory

The pure water contact angle of the pristine PVDF membrane was 82.82°, as indicated in Table 4. It had a high hydrophobicity. The pure water contact angle was drastically lowered after the modification. The pure water contact angle was lowered to 43.34° for the pDA + KH550 modified membrane, in particular. When functional groups such as hydroxyl, carboxyl, and carbonyl groups are present in the molecular structure, substances have negative zeta potentials [37,38]. The zeta potential on the modified membrane surface was more negative because a significant amount of hydroxyl groups were introduced on the membrane surface after the hydrolysis of dopamine and KH550.

Based on the dates in Table 4, Equations (9)–(15) were used to calculate the surface tension and the interfacial energy between the membrane and pectin. The results are, respectively, shown in Table 5 and Table 6. The total surface tension of the two modified membrane surfaces and each sub-item of tension, were significantly increased. $\gamma^{+}$, $\gamma^{-}$, $\gamma^{LW}$ represent the polarity characteristics of the surface, and $\gamma^{LW}$ represents the non-polar characteristics. The polar characteristics of the modified membrane differed significantly from those of the pristine membrane, as shown in Table 5. Therefore, these parameters can be used as a reliable index to evaluate the result of membrane modification. Interfacial energy between pectin and membrane is listed in Table 6. The values of $∆G_{mlp}^{EL}$ and $∆G_{mlp}^{LW}$ have no evident changes, and are not described in detail in the section. $∆G_{mlp\;}^{AB}$ is the main sub-items of $∆G_{mlp}^{TOT}$. After the pDA was changed, the value decreased significantly, changing from a negative number to a positive number when KH550 was added. This means that when KH550 was added, the mutual attraction between the modified membrane and the pectin molecule was weakened, and the interfacial energy between the membrane and the pectin molecules changed from mutual attraction to mutual repulsion.

Figure 10 shows profiles of the interactions between the membrane and sludge foulants in the scenario of two infinite planar surfaces for the control and coated membranes. Negative values indicated mutual attraction, and positive values indicated mutual exclusion. As shown in the figure, the LW and AB interactions were attractive, while the EL interaction was repulsive at a certain distance (0.158 nm, the minimum equilibrium distance between two substances). The U^LW^ curve of the control and coated membranes showed a similar profile and position. There was an obvious discrepancy for U^EL^ and U^AB^ in terms of the profile and position. The U^AB^ curve varied simultaneously with the U^TOT^ curve, indicating that AB and EL together affected the total energy of action. AB interaction played a predominant role in the process of the interaction of the pectin molecule with the membrane. The pDA + KH550-modified membrane showed higher AB interactions than the pristine membrane and the pDA-modified membrane.

Figure 10d illustrates the trend of the total interaction energy with separation distance. The energy was first increased to a constant value (energy barrier) and then continuously decreased to near zero. It can be inferred that the process of pectin molecules approaching the membrane surface underwent two stages: the approach to the membrane surface by overcoming repulsive interaction force, and the approach to the membrane surface with the attractive interaction force. This means that the eventual adhesion of a foulant particle on the membrane surface should overcome a repulsive energy barrier [39], whereby the higher the energy barrier, the more difficult the adhesion of a particle becomes. The total interaction profiles for the control and the two coated membranes are shown in Figure 10d. The energy barrier for the pDA and KH550-coated membrane was 3572.58 kT, prominently higher than the value of 207.67 kT for the pDA-coated membrane and the value of 144.21 kT for the control membrane. Therefore, in terms of the interaction energy and energy barrier, the reduced fouling of the coated membrane in the experiments was well explained.

## 4. Conclusions

The surface of a PVDF membrane was successfully modified by the combination of self-polymerized polydopamine and subsequent hydrolysis of a silane coupling agent. This formed a super-hydrophilic hybrid coating on the surface of the membrane that greatly enhanced its anti-fouling performance and separation performance. The introduction of KH550 solved the problems of roughness with dopamine-modified film and the poor stability of coatings, and provided much better anti-pollution properties than those of dopamine-modified films, due to the formation of hydrophilic coatings that contained Si-OH and Si-O-Si. Combining XDLVO theory and the results of characterization analysis and flux analysis of pure and pectin, the pollution mechanism of the membranes was comprehensively analyzed. For modified membranes, the stronger total energy of the interface interaction and the energy barrier indicated that there was more resistance to mutual adsorption. In summary, the proposed strategy easily and effectively used self-polymerized polydopamine and the subsequent hydrolysis of KH550 to modify the PVDF membrane. The introduction of KH550 optimized the properties of anti-fouling and separation and mitigated problems with pDA-modified membranes, such as poor stability in acids, alkalis, and organic solvents, weaker hydrophilicity, time-consuming modification, and the roughness and unevenness of the coating. These problems affect the separation and purification of the membranes of TMC extracts, because larger and thicker polydopamine particles clog the pores.

## Figures and Tables

**Figure 1 membranes-12-00740-f001:**
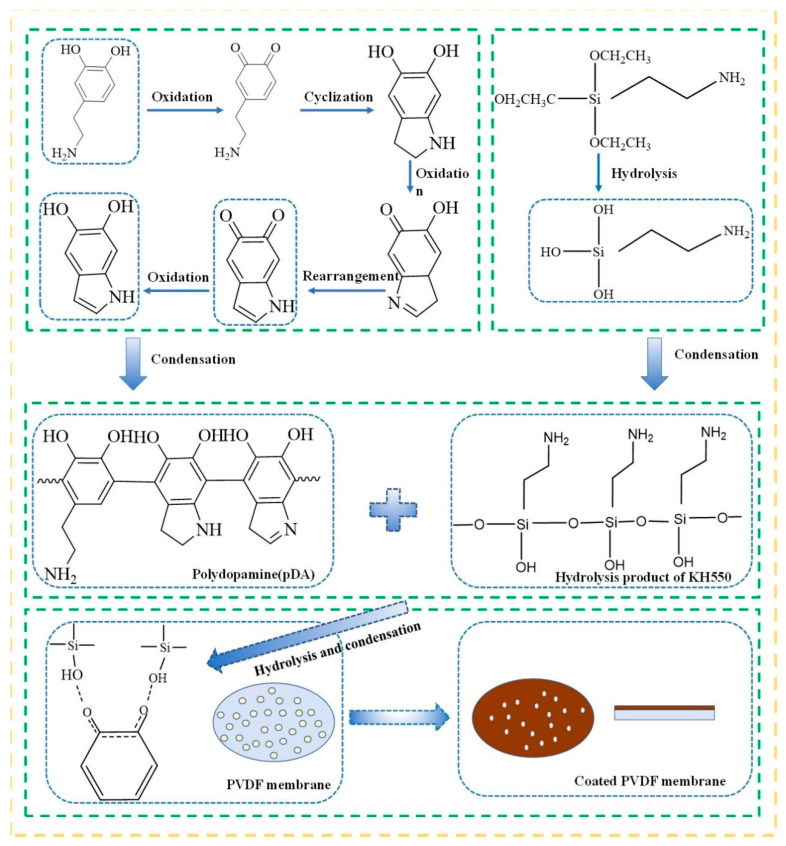
Schematic illustration of the immobilization of pDA/KH550 coatings onto the fiber surface.

**Figure 2 membranes-12-00740-f002:**
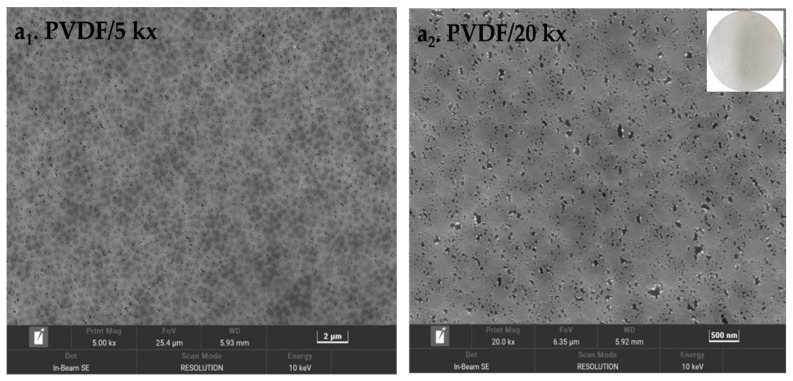
The SEM and pictures of the top surface morphologies of pristine and coated membranes (**a_1_**–**c_1_**): low-resolution images (5 kx), (**a_2_**–**c_2_**): high-resolution images (20 kx)).

**Figure 3 membranes-12-00740-f003:**
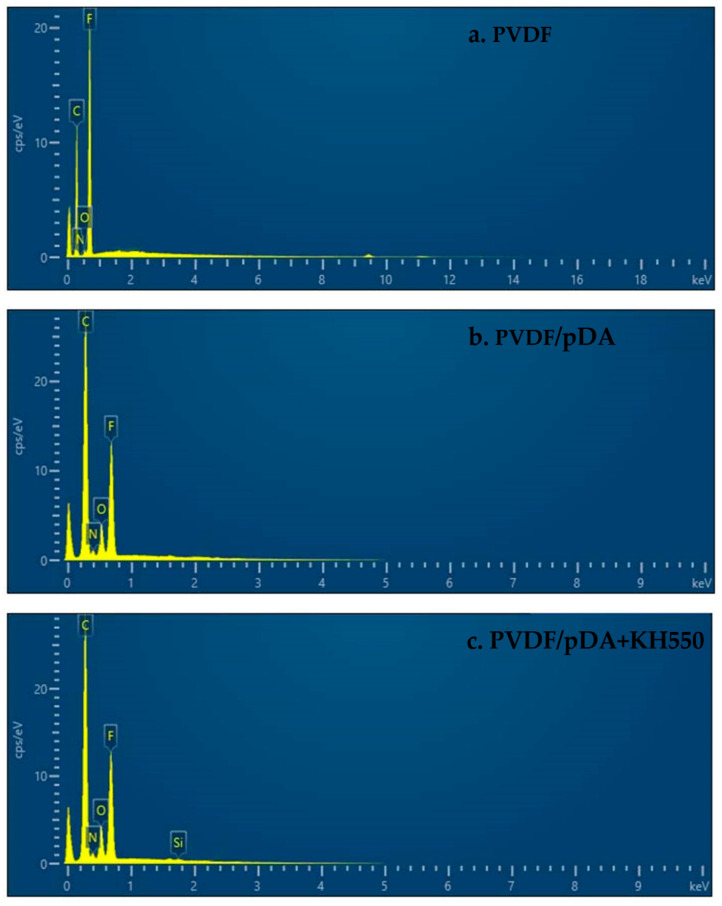
The EDS of elements of pristine and coated membrane surfaces.

**Figure 4 membranes-12-00740-f004:**
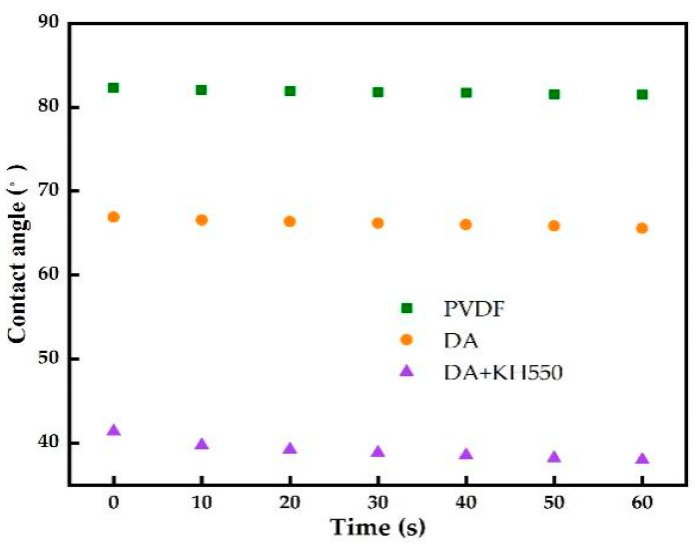
Comparison of the water contact angle of pristine PVDF and coated membranes.

**Figure 5 membranes-12-00740-f005:**
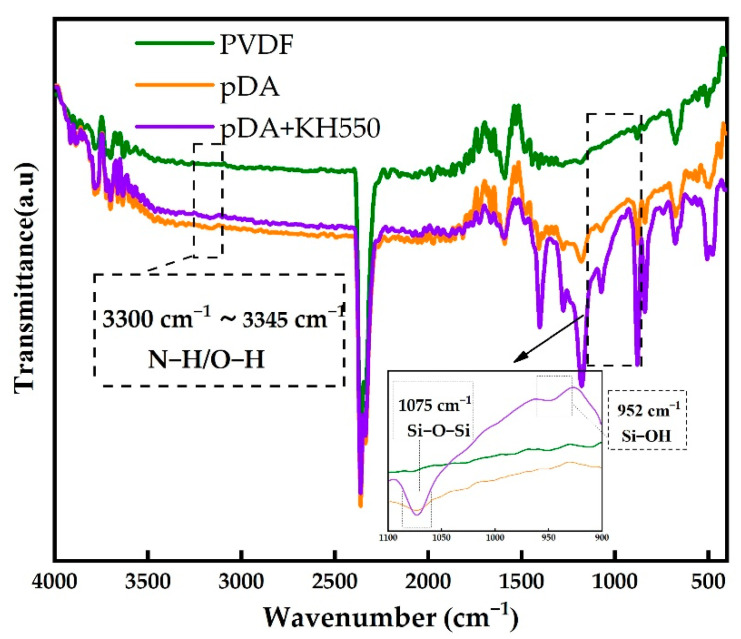
ATR-FTIR spectra of pristine PVDF and coated membranes.

**Figure 6 membranes-12-00740-f006:**
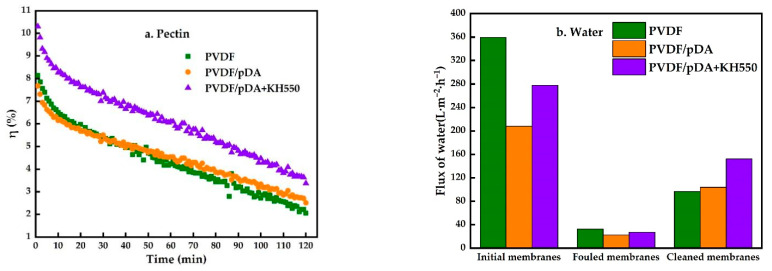
Relative flux of pectin solution (**a**), and flux of water at different stages (**b**).

**Figure 7 membranes-12-00740-f007:**
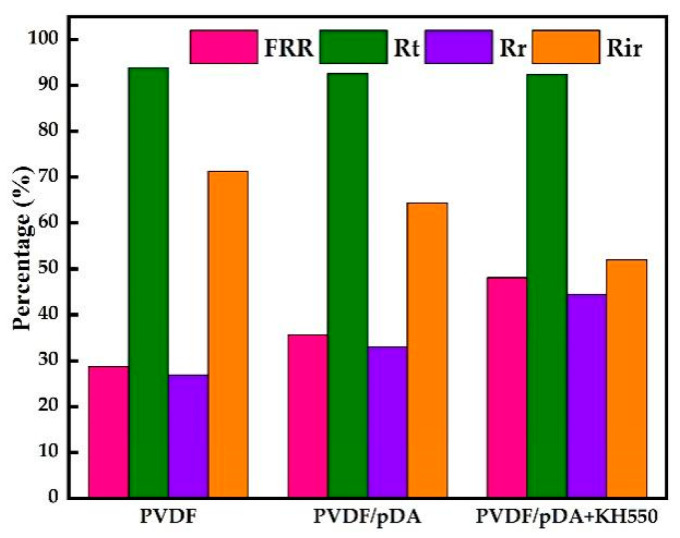
Comparison of fouling index under pectin solution UF test.

**Figure 8 membranes-12-00740-f008:**
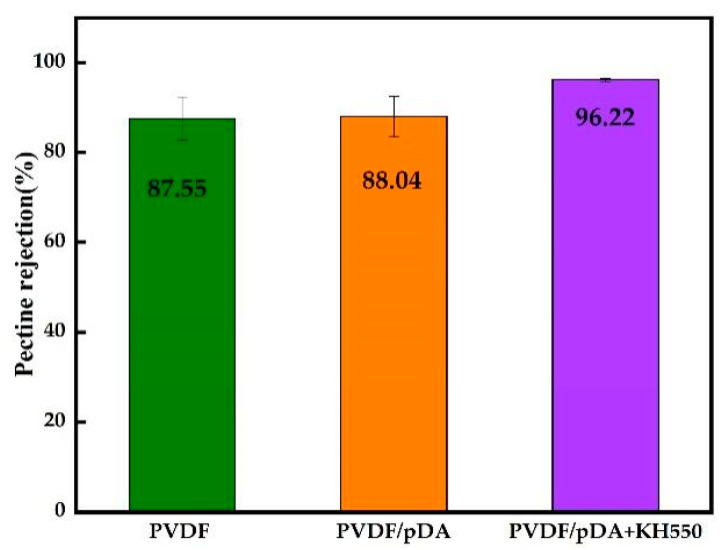
Pectin rejection by the pristine and coated membranes under pectin solution.

**Figure 9 membranes-12-00740-f009:**
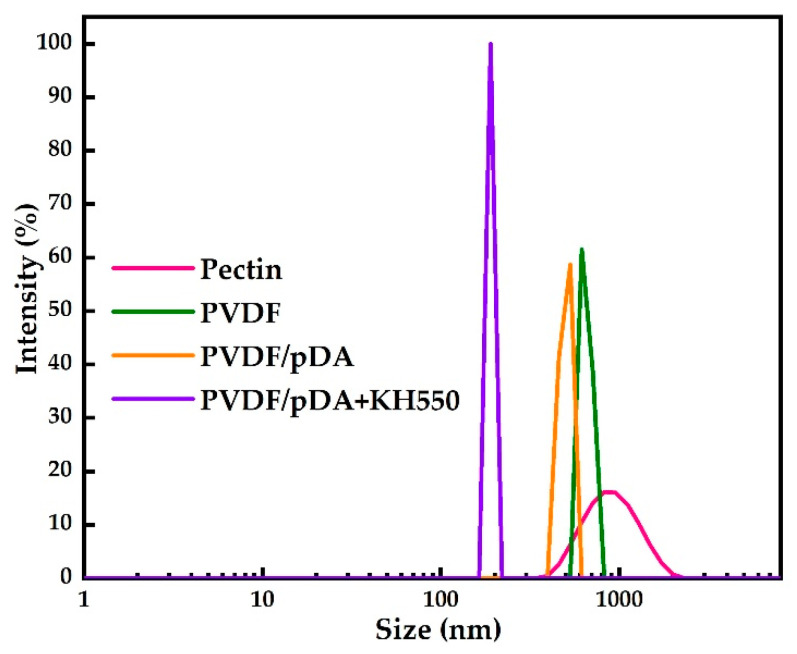
Particle size distribution of the pectin solution and penetrants.

**Figure 10 membranes-12-00740-f010:**
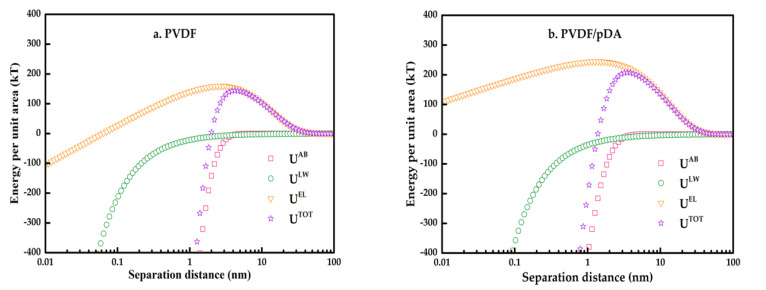
Profiles of specific interactions between the pristine and coated membranes, and the contaminant, as a function of the separation distance (ns: $1\;\mathrm{kT}=4.115\times 10^{-21}\;J$).

**Table 1 membranes-12-00740-t001:** The pure water flux and contact angle of different modified membranes reported in the literature.

Modified Membrane	Sample	Pure Water Flux (L·m^−2^·h^−1^)	Contact Angle (°)	References
HF-PVDF/pDA	HF-PVDF	5670 (30 kPa)	87.50	[21]
	HF-PVDF/pDA	637 (30 kPa)	43.60	
CFs/PDA + TEOS	CFs	/	71.50	[17]
	CFs/PDA + TEOS	/	41.89	
PES/pDA + PEI	PES	45 (100 kPa)	70.00	[20]
	PES/pDA + PEI	55 (100 kPa)	50.00	
PVDF/pDA + TiO_2_	PVDF	135 (100 kPa)	80.10	[15]
	PVDF/pDA + TiO_2_	225 (100 kPa)	47.05	
PSF/TiO_2_ + pDA + KH550	PSF	548 (100 kPa)	66.50	[22]
	PSF/TiO_2_ + pDA + KH550	275 (100 kPa)	35.30	

ns: HF, hollow fiber; CFs, carbon fibers; PSF, polysulfone film.

**Table 2 membranes-12-00740-t002:** The weight of elements on the surface of pristine and modified membranes.

Element	Weight (%)		
	PVDF	PVDF/pDA	PVDF/pDA + KH550
C	60.59	62.13	62.33
N	/	3.05	3.30
O	1.03	7.08	7.49
F	38.37	27.24	26.37
Si	/	/	0.51

**Table 3 membranes-12-00740-t003:** Porosity and mean pore size of pristine and modified membranes.

Sample	Porosity (%)	Mean Pore Size (nm)
PVDF	53.11	40.18
PVDF/pDA	47.02	38.69
PVDF/pDA + KH550	43.69	27.25

**Table 4 membranes-12-00740-t004:** Contact angle and zeta potentials of membranes and pectin.

Sample	$\theta_{w}/\left({}^{\circ}\right)$	θ $g_{/}\left({}^{\circ}\right)$	$\theta_{d}/\left({}^{\circ}\right)$	ζ/(mV)
PVDF	82.82 ± 0.99	78.22 ± 1.61	50.42 ± 1.73	−22.51 ± 0.14
PVDF/pDA	74.57 ± 2.68	59.38 ± 1.23	31.30 ± 5.10	−28.51 ± 5.66
PVDF/pDA + KH550	43.34 ± 5.05	36.80 ± 2.47	23.51 ± 1.95	−31.45 ± 1.93
Pectin	58.74 ± 4.20	59.10 ± 2.40	41.51 ± 2.07	−48.93 ± 2.05

ns: $\theta_{w}$,
θ
_g_,
$\theta_{d}$ is the contact angles of pure water, glycerin, and diiodomethane, respectively.

**Table 5 membranes-12-00740-t005:** Surface tension of membranes and pectin.

Sample	γ/(mJ·m^−2^)				
	$\gamma^{+}$	$\gamma^{-}$	$\gamma^{LW}$	$\gamma^{AB}$	$\gamma^{TOT}$
PVDF	0.0220	8.2154	34.0387	0.1689	33.0717
PVDF/pDA	0.4425	5.2811	44.0964	3.0573	47.1537
PVDF/pDA + KH550	1.1556	23.8494	46.6695	10.4994	57.1689
Pectin	0.1091	23.5819	38.8412	3.2082	42.0495

**Table 6 membranes-12-00740-t006:** Interfacial energy between pectin and membranes.

Sample	$∆G_{mlp}/(\mathrm{mJ}\cdot m^{-2})$			
	$∆G_{mlp}^{LW}$	$∆G_{mlp}^{AB}$	$∆G_{mlp}^{EL}$	$∆G_{mlp}^{TOT}$
PVDF	−3.6230	−21.7788	−0.0007	−25.4225
PVDF/pDA	−6.1637	−11.7145	−0.0009	−17.8791
PVDF/pDA + KH550	−6.7608	24.2249	−0.0002	17.4639

## Data Availability

The data presented in this study are available on request from the corresponding author.

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
