# Peer review of "Anti-Pectin Fouling Performance of Dopamine and (3-Aminopropy) Triethoxysilane-Coated PVDF Ultrafiltration Membrane"

_membranes, 2022, doi:10.3390/membranes12080740_

Round 1

Reviewer 1 Report

Paper requires strong revision.

Comments:

1. MINOR COMMENT - "Membrane technology is a new separation technique." Please revise, membrane technology is now centuries old - for instance, the first reported membrane process is acient greek, using pork intestine for water purification purposes.

2. MINOR COMMENT - "Before modifying the membrane, the PVDF ultrafiltration membrane needs to be pre-treated". Please specify according to what protocol (general, case study) or if home developed as optimal.

3. MINOR COMMENT - Line 149, ? is the pure water flux: the equation is general, J is permeate flux. If authors intend initial pure water flux they should write Jw0(?)

4. MAJOR COMMENT - Figure 6: "However, the flux attenuation of the control membrane is more noticeable than that of the coated membranes." The statement is true but on the other hand it can be seen that in the given time interval all three fluxes of the different membranes tend to a same asymptote. In other words, it seems that all membranes share the same final permeate flux value: if this is the case, most probably a threshold flux is present and if this value is the same to all membranes, the flux decline is given by operating conditions and not by a more/less fouling of the same. Authors should focus on this aspect and analyze data more precisely with insight to fouling indexes, operating conditions and threshold/boundary fluxes since only 30 min of operation were performed. A longer operation would surely have addressed this issue more correctly.

5. MAJOR COMMENT - Figure 9: The figure may be connected to fouling only if pore size of the membranes are provided. Indeed, fouling molecules are those in the range of the pore size of the membrane: if smaller they will pass, if much bigger they will roll over the surface not sticking in the pores. Please provide the relevant data.

6. MAJOR COMMENT - "There is an energy barrier when a foulant particle approaches the membrane surface." All this sectionshould be revised in english. Moreover, authors do strong statements on what they suppose to be true. In fact, they may have a point here; but "energy barriers" or in other words, donnan effects, have also some backdraw. The topic requires to be exploited in a more consistent way.

Author Response

Thank you for your valuable comments, below  is my responses to your suggestions and doubts. Please see the attachment for the resport  and revised manuscript. Please review again. Looking forward to your reply.

1.  This sentence is modified as " Membrane technology is an efficient separation technique. " The details are shown in line 42.

2. The pretreatment operation used in this study refers to the relevant literatures,which have been listed in the revised manuscript (line 117). The purpose of soaking the membrane in pure water is to remove glycerol, which is a protective agent used in membrane storage. In addition, it was found that the highly wet membrane is more conducive to polydopamine adhering to the membrane surface, so it was soaked with ethanol to ensure that the membrane to be modified is sufficiently moistened.

3. It is very sorry for our negligence. The equation in line 165 of the original manuscript is a general formula, J is the permeate flux. Jw0, Jw1, Jw2, and Jf are all calculated using this formula. It has been revised in the revised manuscript (line 167).

4. As you said, it seems that all membranes share the same final permeate flux value. The reason may be that only 30 minutes of operations were performed. According to your suggestion, adjust the final operation time of stage C to 90 minutes. The result is shown in Figure 6 (b). After 90 minutes, the flux of the pristine membrane continued to decline whereas the modified membrane gradually stabilized.

5. Data on mean pore radius and porosity are supplemented in Table 3 of the revised manuscript. Combined with Table 3 and Figure 9, the relationship between particle size, membrane pore size, and membrane fouling was comprehensively analyzed in the revised manuscript. The details are shown in lines 142-151 and 322-323.

6. The formation of the cake layer on the membrane surface is a dynamic process. Hydrodynamic and thermodynamic prop coordinate to determine the adhesion of foulants to the membrane surface. The hydrodynamic drag forces bring the foulants close to the membrane surface, and the thermodynamic forces cause the binding of the foulants to the membrane and eventually form a cake layer. Therefore, assessment of interfacial energy and energy barriers between the foulants and membrane surface would be very conducive to a better understanding of membrane fouling. Inappropriate formulations and grammar have been corrected in the text. The details are shown in lines 364-370.

Kind regards,
Ms. Dengrong Lu

Reviewer 2 Report

The manuscript contains interesting research material but requires completion and correction.

My comments and observations:

1.      Title - should be modified, because in the presented form it is not known whether pectin causes fouling or counteracts fouling

2.      Abstract - the presented achievements can be supported by the obtained numerical results, e.g .: flux increased (line 28), total interface (line 32-33).

3.      Keywords - keywords should include the word membrane or ultrafiltration

4.       Introductions, line 64-66 – PVDF is a synthetic polymer

5.      2.2. Membrane Preparation - please provide details of the membrane and membrane filtration, i.e. membrane type (flat, tubular, active surface of the tested membrane, type of filtration system: dead-end, cross-flow, etc.)

6.      2.4. Filtration Experiments - why a higher pressure was used to condition the membrane (300 kPa) than in the treatment of water (200 kPa)?

7.      2.5. XDLVO - expand on this abbreviation

8.      Figure 3 - incorrect signature under Fig. 3b

9.      3.2. Analysis of Anti-Fouling and Separation Performance - no description for fig. 6a

10.   Lines 312, 321 - control membrane was written and the previous sections wrote pristine membrane? does control and pristine refer to a different membrane?

Author Response

Dear revierer, 

Thank you for your valuable suggestions, below is my responses for  your suggestions andquestions, Please see the attachment for resport.  Looking forward to your reply.  

1. Title - should be modified, because in the presented form it is not known whether pectin causes fouling or counteracts fouling

Response:  The title has been modified as " Anti-pectin Fouling Performance of Dopamine and (3-aminopropy) Triethoxysilane-Coated PVDF Ultrafiltration Membrane".

2. Abstract - the presented achievements can be supported by the obtained numerical results, e.g .: flux increased (line 28), total interface (line 32-33).

Response:  The obtained numerical results have been supplied in the revised manuscript.

3. Keywords - keywords should include the word membrane or ultrafiltration

Response: The keyword "ultrafiltration membrane" has been supplied in the revised manuscript.

4. Introductions, line 64-66 – PVDF is a synthetic polymer

Response:This sentence has been modified in the revised manuscript (line 67).

5. Membrane Preparation - please provide details of the membrane and membrane filtration, i.e. membrane type (flat, tubular, active surface of the tested membrane, type of filtration system: dead-end, cross-flow, etc.)

Response:  The details of the membrane type (line 112-113) and membrane filtration system (line 154-155) are supplied in the revised manuscript.

6. Filtration Experiments - why a higher pressure was used to condition the membrane (300 kPa) than in the treatment of water (200 kPa)?

Response: Higher pressure allows the membrane to reach a balanced state (membrane compaction) faster to reduce individual differences of flux in the initial stages of membrane filtration. Therefore, a higher pressure (300 kPa) is used to condition the membrane.

7. XDLVO - expand on this abbreviation

Response:  The full name of the XDLVO theory has been supplied in the line 101-102 of the abstract in the revised manuscript.

8.Figure 3 - incorrect signature under Fig. 3b

Response:  We merged Figure 3 with Figure 2 in the revised manuscript.

9. Analysis of Anti-Fouling and Separation Performance - no description for fig. 6a

Response: Thanks for the very kind suggestion. The description of Figure 6a has been supplied in the revised manuscript (lines 282-287).

10. Lines 312, 321 - control membrane was written and the previous sections wrote pristine membrane? does control and pristine refer to a different membrane?

Response: It is very sorry for our negligence. Control and pristine refer to the same membrane. It has been uniformly written as the pristine membrane in the revised manuscript.

Kind regards,
Ms. Dengrong Lu

Reviewer 3 Report

Major Revision

The author has demonstrated the feasibility of anti-Fouling performance of dopamine and (3- aminopropy) Triethoxysilane-Coated Polyvinylidene Fluoride ultrafiltration membrane.  Overall, this is a decent research topic which could be noteworthy for researchers’ studying membrane fabrication for antifouling ultrafiltration applications. However, the presentation of the article should be modified for better understanding and readability. There are few issues which were pointed out.

Specific Comments

1.     The title can be simplified for better understanding and readability. There are lot of grammatical typos and spelling errors in the manuscript. Please check thoroughly. For instance, “Triethoxysilane” in title.

2.     Next, introduction section needs minor revision. There are some research works that have been already executed based on antifouling UF membrane. The state-of-art is missing from the manuscript. The author should have added the previous research outputs (conditions/parameter/overall results) by comparing the present one in tabular form to show the viability of the present study.

3.     Coming to the next point, the Figure 2 must be well explained with high scientific discussion. The observation must be validated with appropriate reference for better insight.

4.     What we can learn from Figure 3? Are we gaining any scientific value? Kindly remove it or merge it with Figure 2.

5.     The captions of Figure 4 should be changed. Since it’s a graphical representation. More scientific discussion is needed for the Figure 4, “The introduction of the functional group greatly enhanced the hydrophilicity and permeability of the membrane surface.” Please comment on this with valid references.  

6.     While dealing with membrane research topic, the authors are encouraged to provide following data (if possible):

(a)   Thickness of virgin and modified membrane

(b)   In section 2.3, EDS has been suggested by the author, however, the EDS data is missing from the manuscript.

(c)   The cross-section view of virgin and modified membrane

Author Response

Dear reviewer,

Thank you for your valuable suggestions. Below is my responses for your suggestions and questions, please see the attachment for the resport. Looking forward to your reply.

1. implified for better understanding and readability. There are lot of grammatical typos and spelling errors in the manuscript. Please check thoroughly. For instance, “Triethoxysilane” in title.

Response:  The title has been modified as " Anti-pectin Fouling Performance of Dopamine and (3-aminopropy) Triethoxysilane-Coated PVDF Ultrafiltration Membrane".

2. Next, introduction section needs minor revision. There are some research works that have been already executed based on antifouling UF membrane. The state-of-art is missing from the manuscript. The author should have added the previous research outputs (conditions/parameter/overall results) by comparing the present one in tabular form to show the viability of the present

Response: Some of the anti-fouling methods and results in the references have been supplied in Table 1 in the revised manuscript.

3. Coming to the next point, the Figure 2 must be well explained with high scientific discussion. The observation must be validated with appropriate reference for better insight.

Response: Figure 2 has been well explained in the revised manuscript and the appropriate references have also been supplied. The details are shown in lines (228-235).

4. What we can learn from Figure 3? Are we gaining any scientific value? Kindly remove it or merge it with Figure 2.

Response: Figure 3 in the original manuscript illustrates the color change of the membrane surface after modification. The degree of dopamine polymerization can be visually observed by the color of the surface; the darker the color, the more serious the degree of polymerization. In order to make the content more concise, Figure 3 in the original manuscript has been merged with Figure 2.

5. The captions of Figure 4 should be changed. Since it’s a graphical representation. More scientific discussion is needed for the Figure 4, “The introduction of the functional group greatly enhanced the hydrophilicity and permeability of the membrane surface.” Please comment on this with valid references.  

Response:The caption of Figure 4 has been changed to " Comparison of the water contact angle of pristine PVDF and coated membranes." Combine the result of ATR-FTIR and literatures, this section has been revised in revised manuscript.

6. While dealing with membrane research topic, the authors are encouraged to provide following data (if possible):

(a)   Thickness of virgin and modified membrane

(b)   In section 2.3, EDS has been suggested by the author, however, the

(c)   The cross-section view of virgin and modified membrane

Response: Supplementary experiments and data analysis for EDS has been supplied in the revised manuscript. The details are shown in Table 2 and Figure 3.

Round 2

Reviewer 1 Report

Main issues were addressed.

Only this comment hold on:

4. MAJOR COMMENT - Figure 6: "However, the flux attenuation of the control membrane is more noticeable than that of the coated membranes." The statement is true but on the other hand it can be seen that in the given time interval all three fluxes of the different membranes tend to a same asymptote. In other words, it seems that all membranes share the same final permeate flux value: if this is the case, most probably a threshold flux is present and if this value is the same to all membranes, the flux decline is given by operating conditions and not by a more/less fouling of the same. Authors should focus on this aspect and analyze data more precisely with insight to fouling indexes, operating conditions and threshold/boundary fluxes since only 30 min of operation were performed. A longer operation would surely have addressed this issue more correctly.

The changes made to the text in this revision do not tackle the problem and result insufficient.

Author Response

Dear reviewer,

Thank for your suggestions again. We have studied this section carefully and have made corrections in revised manuscript. The following is the response to your suggestion about the manuscript. Please see the attachment about the resport. Looking forward to your reply.

MAJOR COMMENT - Figure 6: "However, the flux attenuation of the control membrane is more noticeable than that of the coated membranes." The statement is true but on the other hand it can be seen that in the given time interval all three fluxes of the different membranes tend to a same asymptote. In other words, it seems that all membranes share the same final permeate flux value: if this is the case, most probably a threshold flux is present and if this value is the same to all membranes, the flux decline is given by operating conditions and not by a moreless fouling of the same. Authors should focus on this aspect and analyze data more precisely with insight to fouling indexes, operating conditions and threshold/boundary fluxes since only 30 min of operation were performed. A longer operation would surely have addressed this issue more correctly.

Response: Thanks for your suggestions again about this section. We have studied Figure 6(b) again carefully and find it is inappropriate for taking time as the abscissa. The data in Figure 6(b) is not a continuous 180-minute experimental process. Figure 6b is the pure water permeate flux of different membranes in various filtration stages as described in “2.4. Filtration Experiments”, including the water flux of initial membranes (JW0), fouled membranes (JW1), and cleaned membranes (JW2). We are very sorry for the inappropriate drawing in the original manuscript. We have amended it in the revised manuscript. The details are shown in lines 282-302.

Kind regards,

Ms. Dengrong Lu

Reviewer 3 Report

The authors have addressed all my queries with high scientific discussions. Therefore, the revised manuscript can be accepted in the present format. 

Author Response

Dear reviewer,

Thanks for you suggestions again. We will further improve the article and hope to meet your  magazine's requirements.

Kind regards,
Ms. Dengrong Lu

Round 3

Reviewer 1 Report

Authors have addressed all issues and paper can be published.